# OpenReview forum: "DoLa: Decoding by Contrasting Layers Improves Factuality in Large Language Models"
_ICLR.cc/2024/Conference — ICLR 2024 poster_

### Official Review · Reviewer_Cc1D · 2023-10-27

**Soundness:** 3 good
**Presentation:** 3 good
**Contribution:** 4 excellent
**Rating:** 8
**Confidence:** 4

**Summary:**

This paper proposes to improve the factuality of generative LLM by contrasting the differences in logits of different transformer layers. This is based on the analytic results that when predicting important named entity words, the final layer of LLM tend to be very different from some early layers. Therefore, the authors propose to subtract the logits obtained from early layers from the final layer to reduce hallucination.

**Strengths:**

1. the proposed method seems to be quite effective at improving the factuality QA metrics
2. the paper presents good analysis that motivates the method
3. the paper also has good ablations of the experimental results that can be helpful for future work.

**Weaknesses:**

1. the proposed method likely decreases the inference speed, but there is little discussion on how much it slows down the decoding.

**Questions:**

1. subtracting early layer logits make sense, but this might also be related to residual connection from early layers to final layer. Have you tried just removing the residual connection between the premature layer and final layer? Would that be helpful without too much decrease in inference speed?
2.  what's the inference speed of your method compared to the baseline?

---

> ### Author Response · Authors · 2023-11-17
> **Author response to Reviewer Cc1D**
>
> We thank Reviewer Cc1D for the constructive comments!
>
> > subtracting early layer logits make sense, but this might also be related to residual connection from early layers to final layer. Have you tried just removing the residual connection between the premature layer and final layer? Would that be helpful without too much decrease in inference speed?
>
> Removing the residual connections of the premature layer could potentially be another method to downplay the early layers and emphasize the higher layers, without causing any latency. We quickly tried some initial experiments with this idea. However, we found that once we remove the residual connection from one of the attention layers or MLP layers, the decoding output will become a sequence of repeated tokens. For example, when removing the MLP residual connection of layer 12, we got:
>
> `Question: Eliza's rate per hour for the first 40 hours she works each week is $10. She also receives an overtime pay of 1.2 times her regular hourly rate. If Eliza worked for 45 hours this week, how much are her earnings for this week?`
>
> `Model Output: s that that that that s that s s s that that s that that s that that s s s that that that s that that that that s that that that that that that …`
>
> We suspect that the removal of residual connections causes a significant distribution shift for the hidden states, thus resulting in out-of-distribution inputs for the next layer and subsequently unusual model behavior. It seems more difficult to manipulate the continuous hidden states, compared to manipulating only the output logits as we did in DoLa, as the output logits over discrete vocabularies are more explainable. However, we believe that this could be a direction that is worth exploring more carefully in the future. Additional fine-tuning may be required to solve the distribution shift issue.
>
> > what's the inference speed of your method compared to the baseline?
>
> For the latency analysis, we use the 817 examples from TruthfulQA with the default 6-shot in-context demonstration prompt which has an average input length is 250.3 after concatenating the prompt with the questions. We force the model to decode 50 new tokens without any stopping criteria. All the experiments were run on the machine with NVIDIA 32G V100s equipped with 40-core CPUs of Intel(R) Xeon(R) Platinum 8168 CPU @ 2.70GHZ. We run the models with 16-bit floating point and batch size = 1. For 7b/13b/33b/65b models, we use 1/2/4/8 V100 GPUs, respectively. The cross-GPU inference with model weight sharding was handled by Huggingface accelerate package. [1]
>
> Here is the latency (ms/token) and throughput (tokens/sec) for 4 models.
>
> Latency:
> | Model | Baseline (ms/token) | DoLa (ms/token) | Ratio of DoLa/Baseline |
> |-------|---------------------|-----------------|-----------------|
> | 7B    | 45.4                | 48.0            | ×1.06           |
> | 13B   | 77.3                | 83.1            | ×1.08           |
> | 33B   | 146.7               | 156.7           | ×1.07           |
> | 65B   | 321.6               | 324.9           | ×1.01           |
>
> Throughput:
> | Model | Baseline (tokens/sec) | DoLa (tokens/sec) | Ratio of DoLa/Baseline |
> |-------|-----------------------|-------------------|------------------------|
> | 7B    | 22.03                 | 20.83             | ×0.95                  |
> | 13B   | 12.94                 | 12.03             | ×0.93                  |
> | 33B   | 6.82                  | 6.38              | ×0.94                  |
> | 65B   | 3.11                  | 3.08              | ×0.99                  |
>
> We can see that the speed is slightly slower than the vanilla decoding baseline by 1%~8%, which is in an acceptable range. This result suggests that DoLa is a practical decoding strategy that has the potential to be applied to real-world applications.
>
> [1] https://huggingface.co/docs/accelerate/concept_guides/big_model_inference

---

> ### Author Response · Authors · 2023-11-21
> **Feedback on our rebuttal**
>
> Thank you once again for your insightful review! Since the discussion period will end in less than two days, we wanted to make sure that we have adequately addressed the issues you raised. We also hope that our additional experiments have addressed most of your concerns. We would appreciate your feedback on our responses!

---

### Official Review · Reviewer_VLmJ · 2023-10-28

**Soundness:** 4 excellent
**Presentation:** 4 excellent
**Contribution:** 3 good
**Rating:** 8
**Confidence:** 4

**Summary:**

This paper proposes a new decoding scheme for language models that contracts the final layers’ outputs with the middle layers’. At each decoding step, the model produces a distribution over the vocab after every layer; the one that is most different from the final layer is selected, measured by Jensen-Shannon divergence. It then contrasts the final layer’s logits with the selected layer’s, by subtracting the latter from the former before taking a softmax. This method is dubbed DoLA.

DoLA is, in many sense, similar to contrastive decoding [1], but uses the same model’s middle layer instead of a separate smaller model as the “weaker model.” DoLA dynamically selects the middle layer to contrast with at each time step. DoLA has a static variant, where the middle layer is selected based on performance on the validation data and fixed for all instances and time steps. The assumption behind DoLA is that factual knowledge is injected into the model at higher layers, and thus downplaying the lower layers’ decision can help improve factuality. To support this, some empirical (but anecdotal) evidence is presented.

DoLA is evaluated on various downstream tasks, including TruthfulQA, Factor, strQA, and GSM8K. Results show that DoLA, when used with Lllama and MPT, improves the performance on all datasets, and outperforms contrastive decoding on almost all datasets with Llama. Analysis shows that DoLA only incurs minimal overhead in the model’s latency.

Overall, I find this paper an interesting read. I’m leaning negative due to various issues that I raised below. I’m happy to revisit the score if the authors can address my concerns.

[1] https://arxiv.org/abs/2210.15097

**Strengths:**

- DoLA is a simple method and can be used with many open-source models out of the box.
- Strong and consistent improvements in a variety of tasks
- The finding that the final layer’s distribution is very similar to many middle layers’ is novel and interesting to me
- Presentation is very clear

**Weaknesses:**

- The hypothesis behind DoLA that factual knowledge is injected at top layers is not convincing and only weakly supported
- The evidence in Figure 2 is anecdotal; I wonder whether there are quantitative results to back it up
- The efficiency results and claims can be further elaborated (see questions below)
- I don’t see a way to apply DoLA to proprietary models.
- [Important] I suspect that DoLA will reduce back to contrasting with the 0-th layer in practice (see questions below)
- It would be great to investigate DoLA’s impact on the language model’s generation quality

**Questions:**

- [Important] In the example in Figure 2, almost all layers selected by DoLA will be the wording embedding layer. In such cases, assuming $q_0(x_t)$ has a substantial amount of mass on $x_t$, DoLA is essentially discouraging the model from generating the same token as the previous one. Maybe I’m missing something obvious, but I am not entirely convinced by the “factuality happens at higher layers” narrative.
- [Important] Following the above, have the authors compared to a baseline that always selects the 0-th layer? I won’t be surprised if it achieves strong performance, since it is doing exactly the same thing as DoLA on most generation steps.
- Figure 2 is nice; I wonder whether the authors have any quantitative results on this.
- Decoding from every layer seems expensive. I’m surprised by the efficiency results presented in Section 4.4. Can the authors provide more details on, e.g., how is this measured and on what tasks/hardware, and the generation lengths?
- [Important] Following the above, latency is only one aspect of efficiency and a determining factor of whether DoLA “can be widely applied with negligible cost.” It would be great if the authors can quantify DoLA’s impact on throughput and memory overhead too.

---

> ### Author Response · Authors · 2023-11-17
> **Author response to Reviewer VLmJ (1/4)**
>
> We thank Reviewer VLmJ for the constructive comments!
>
> > [Important] In the example in Figure 2, almost all layers selected by DoLA will be the wording embedding layer.
>
> In Figure 2, when predicting important tokens that are related to factual information, the higher part of the layers are more likely to be selected. When predicting the easy-to-predict tokens like function words, the word embedding layer is more likely to be selected. The claim that “almost all layers selected will be the wording embedding layer” only applies to easy-to-predict tokens, not all of the tokens.
>
> > In such cases, assuming $q_0(x_t)$ has a substantial amount of mass on $x_t$, DoLA is essentially discouraging the model from generating the same token as the previous one. Maybe I’m missing something obvious, but I am not entirely convinced by the “factuality happens at higher layers” narrative.
>
> Because LLaMA’s word embedding layer is **NOT** tied with the LM head [1], when feeding the word embedding of the previous token directly into the LM head, the output distribution ($q_0(x_t)$) would not put a substantial amount of mass on the previous token ($x_t$) at all. Instead, the output distribution will be closer to a non-contextual prediction of the next token ($x_{t+1}$) only conditioned on the previous token ($x_t$), similar to the idea of a bi-gram language model ($P(x_t+1|(x_t))$). Thus, **DoLa is NOT discouraging the model from generating the same token as the previous one.** I believe this is a misunderstanding.
>
> > [Important] Following the above, have the authors compared to a baseline that always selects the 0-th layer? I won’t be surprised if it achieves strong performance, since it is doing exactly the same thing as DoLA on most generation steps.
>
> Contrasting with the 0-th layer indeed improves the performance. Following the above discussion, the effects of the 0-th layer prediction will be similar to a bi-gram LM. Contrasting the final predictions with the bi-gram LM prediction should be beneficial, as the bi-gram predictions contain the non-contextual language statistics bias learned from the pretraining data. Contrasting the final layer with the 0-th layer should be able to remove this non-contextual bias from the final contextualized predictions, emphasize the contextual information that is later injected in the middle layers, and thus make the final prediction unbiased and better.
>
> |            |       7b       |               |       |      13b      |               |       |      33b      |               |       |      65b      |               |       |
> |------------|:--------------:|:-------------:|:-----:|:-------------:|:-------------:|:-----:|:-------------:|:-------------:|:-----:|:-------------:|:-------------:|:-----:|
> |            | MC1  | MC2  | MC3 | MC1  | MC2  | MC3 | MC1  | MC2  | MC3 | MC1  | MC2  | MC3 |
> | Vanilla    | 25.6 | 40.6 | 19.2| 28.3 | 43.3 | 20.8| 31.7 | 49.5 | 24.2| 30.8 | 46.9 | 22.7|
> | DoLa - layer 0 | 31.6 | 61.7 | 30.1| 28.5 | 62.3 | 30.2| 31.4 | 61.1 | 31.1| 31.0 | 63.6 | 31.2|
> | DoLa       | 32.2 | 63.8 | 32.1| 28.9 | 64.9 | 34.8| 30.5 | 62.3 | 34.0| 31.1 | 64.6 | 34.3|
>
> In the above table, we show the result of DoLa with only contrast to the 0-th layer. In general, using only the 0-th layer already improves the performance compared to the vanilla decoding baseline. DoLa incorporates information from multiple layers can further increase the scores.
>
> [1] https://github.com/facebookresearch/llama/issues/138; We also have verified this by manually comparing the embedding weights and the LM head weights of LLaMA. They are completely different weights.

---

> ### Author Response · Authors · 2023-11-17
> **Author response to Reviewer VLmJ (2/4)**
>
> > Figure 2 is nice; I wonder whether the authors have any quantitative results on this.
>
> Thanks for the inspiring suggestion! We include an additional quantitative study, where we use the validation set of the CoNLL-2003 name entity recognition dataset [2] with 3.25K examples. We calculate which layer has the largest JS-divergence with the final layer when LLaMA-7B predicts the next token with **teacher forcing** (we simply call this layer the "critical layer" for short). We subdivide the results into two parts by whether LLaMA is predicting an entity token or a non-entity token.
>
> | Layer | Entity Tokens | Non-Entity Tokens |
> |-------|---------------------|-------------------|
> | 0     | 35.56%              | 75.55%            |
> | 2     | 0.05%               | 0.08%             |
> | 4     | 0.94%               | 0.36%             |
> | 6     | 0.94%               | 0.14%             |
> | 8     | 1.05%               | 0.27%             |
> | 10    | 0.05%               | 0.33%             |
> | 12    | 2.10%               | 0.65%             |
> | 14    | 0.00%               | 0.33%             |
> | 16    | 0.00%               | 0.16%             |
> | 18    | 0.00%               | 0.05%             |
> | 20    | 1.69%               | 0.47%             |
> | 22    | 9.69%               | 1.76%             |
> | 24    | 10.38%              | 2.62%             |
> | 26    | 2.08%               | 2.17%             |
> | 28    | 10.06%              | 2.11%             |
> | 30    | 25.40%              | 12.98%            |
>
> We can find that 75% of the time the critical layer will be layer 0 when predicting non-entity tokens. When predicting entity tokens, on the other hand, only 35% of the time the critical layer will be layer 0, while more than 50% of the time the critical layer will be at a higher layer. This experiment can quantitatively support our observations in Figure 2.
>
> Note that we use teacher forcing to send the **ground truth** into LLaMA to predict the next word for each token in the sentence. And the ground truth sentences are not generated by LLaMA. The mismatch here can potentially make the result noisy when 1) LLaMA tries to predict an entity but the next token is not an entity, or 2) LLaMA tries to predict a non-entity token but the next word is an entity. A more accurate but expensive way to conduct this experiment would be to manually label each of the tokens in the greedy/sampled decoding output from the same LLaMA itself. However, from the current experiments we have already seen such a trend in the NER dataset.
>
> [2] Sang, Erik Tjong Kim, and Fien De Meulder. "Introduction to the CoNLL-2003 Shared Task: Language-Independent Named Entity Recognition." In Proceedings of the Seventh Conference on Natural Language Learning at HLT-NAACL 2003, pp. 142-147. 2003. https://huggingface.co/datasets/conll2003

---

> ### Author Response · Authors · 2023-11-17
> **Author response to Reviewer VLmJ (3/4)**
>
> > Decoding from every layer seems expensive. I’m surprised by the efficiency results presented in Section 4.4. Can the authors provide more details on, e.g., how is this measured and on what tasks/hardware, and the generation lengths?
>
> In the latency analysis in Section 4.4, we use the 817 examples from TruthfulQA with the default 6-shot in-context demonstration prompt which has an average input length is 250.3 after concatenating the prompt with the questions. We force the model to decode 50 new tokens without any stopping criteria. One of the reasons we make DoLa so efficient may be the fact that we use batched tensor operations for computing JS-divergence all at once for all the candidate layers together.
>
> All the experiments were run on the machine with NVIDIA 32G V100s equipped with 40-core CPUs of Intel(R) Xeon(R) Platinum 8168 CPU @ 2.70GHZ. We run the models with 16-bit floating point and batch size = 1. For 7b/13b/33b/65b models, we use 1/2/4/8 V100 GPUs, respectively. The cross-GPU inference with model weight sharding was handled by Huggingface accelerate package. [3]
>
> > [Important] Following the above, latency is only one aspect of efficiency and a determining factor of whether DoLA “can be widely applied with negligible cost.” It would be great if the authors can quantify DoLA’s impact on throughput and memory overhead too.
>
> Thanks for the suggestion. We conduct additional experiments to measure throughput and memory overhead.
>
> **Memory Overhead:**
>
> To measure the overhead, we calculate (a) the occupied GPU memory before the first forward pass (b) the peak GPU memory during the forward passes. And then we can compute the memory overhead by (b) - (a), or the proportion of overhead [(b) - (a)] / (a) in %.
> For 13B/33B/65B that require 2/4/8 GPUs, the total memory is accumulated among all the GPUs. The results are shown below:
>
> | Model                                      | LLaMA-7b             |                | LLaMA-13b            |                | LLaMA-30b            |                | LLaMA-65b            |                |
> |---------------------------------------------|----------------------|----------------|----------------------|----------------|----------------------|----------------|----------------------|----------------|
> |                                             | Vanilla              | DoLa           | Vanilla              | DoLa           | Vanilla              | DoLa           | Vanilla              | DoLa           |
> | (a) GPU Memory Before Forward (MB)          | 12916.5              | 12916.5        | 25025.8              | 25025.8        | 55715.7              | 55715.7        | 124682.6             | 124682.6       |
> | (b) Peak GPU Memory During Forward (MB)     | 13233.9              | 13385.7        | 25510.7              | 25674.8        | 57057.5              | 57390.2        | 126950.0             | 127606.8       |
> | (b) - (a) GPU Memory Overhead (MB)          | 317.4                | 469.2          | 484.9                | 681.6          | 1341.9               | 1674.5         | 2267.4               | 2924.3         |
> | [(b) - (a)] / (a) GPU Memory Overhead (%)   | 2.5%                 | 3.6%           | 1.9%                 | 2.7%           | 2.4%                 | 3.0%           | 1.8%                | 2.4%          |
>
> We can see that during the forward pass of LLaMA-7B, the overhead for vanilla decoding is 2.5% while DoLa requires 3.6%. There is only 1.1% difference for the memory overhead between Vanilla and DoLa. For 13b/30b/65b models, the difference is even smaller than 1%. This result shows that the difference in memory overhead between DoLa and the vanilla decoding baseline is still negligible.
>
> **Throughput:**
>
> Here is the throughput (tokens/sec) for 4 models.
>
> | Model | Baseline (tokens/sec) | DoLa (tokens/sec) | Ratio of DoLa/Baseline |
> |-------|-----------------------|-------------------|------------------------|
> | 7B    | 22.03                 | 20.83             | ×0.95                  |
> | 13B   | 12.94                 | 12.03             | ×0.93                  |
> | 33B   | 6.82                  | 6.38              | ×0.94                  |
> | 65B   | 3.11                  | 3.08              | ×0.99                  |
>
>
> We can observe that the differences are generally less than 7%, similar to the extent of increased latency we have shown in Section 4.4.
>
> [3] https://huggingface.co/docs/accelerate/concept_guides/big_model_inference

---

> ### Author Response · Authors · 2023-11-17
> **Author response to Reviewer VLmJ (4/4)**
>
> > Weakness: I don’t see a way to apply DoLA to proprietary models.
>
> DoLa is generally applicable to transformer-based LLMs. The accessibility of blackbox proprietary models like ChatGPT and GPT3/4 is out of the scope discussed in our paper.
>
> We believe that under the scope of academic research, we should encourage research that makes progress in improving or understanding the fundamental mechanism of these large models, instead of considering whether the research favors closed-source proprietary models.
>
> In fact, if all the researchers were encouraged to focus only on methods (e.g. prompting) that work for black box API LLMs, then we would not have a chance to discover the underlying mechanism inside the black box of LLMs, and the possible improvement that can be made would be very limited as we are restricted to surface-level interactions.
>
> > Weakness: It would be great to investigate DoLA’s impact on the language model’s generation quality
>
> Thanks for the insightful suggestion! We conduct an additional study of the quality of generated text using GPT4, given the fact that several prior studies [4][5] have shown the great potential of GPT-4 to serve as an alternative to human evaluation. And the effect is stable over different prompts and instructions. [6]
>
> We adopt the pairwise evaluation code from Vicuna QA. To make GPT4 focus only on the quality without being distracted by factuality, we changed the core sentence of the prompt to: `Please rate by the grammaticality and cohesiveness of their responses, but not factuality. You are not required to verify the factual accuracy of the answers. Each assistant receives an overall score on a scale of 1 to 10, where a higher score indicates better quality.`
>
> By using the prompt above, we observed GPT-4 can rate the answers based on grammaticality and cohesiveness without checking the factual correctness. The results are shown below, where the scores are the average scores from 80 questions in Vicuna QA, on a scale of 1 to 10.
>
> | Model      | Vanilla           | DoLa                    |
> |------------|-------------------|-------------------------|
> | LLaMA-7B   | 6.44              | 6.96                    |
> | LLaMA-13B  | 7.06              | 7.98                    |
> | LLaMA-33B  | 6.89              | 7.84                    |
> | LLaMA-65B  | 8.04              | 8.01                    |
>
> We can observe that for 7B/13B/33B models, DoLa has better grammaticality and cohesiveness compared to the vanilla decoding baseline. For the largest 65B model, DoLa achieves a score that is almost the same as vanilla decoding. We conclude that when evaluating text generation quality without considering factuality, DoLa is still on par with (65B) or better than (7B/13B/33B) vanilla decoding.
>
> [4] Chiang, Cheng-Han, and Hung-yi Lee. "Can Large Language Models Be an Alternative to Human Evaluations?." arXiv preprint arXiv:2305.01937 (2023).
>
> [5] Liu, Yang, Dan Iter, Yichong Xu, Shuohang Wang, Ruochen Xu, and Chenguang Zhu. "G-Eval: NLG Evaluation using GPT-4 with Better Human Alignment." arXiv preprint arXiv:2303.16634 (2023).
>
> [6] Chiang, Cheng-Han, and Hung-yi Lee. "A Closer Look into Automatic Evaluation Using Large Language Models." arXiv preprint arXiv:2310.05657 (2023).

---

> ### Author Response · Authors · 2023-11-21
> **Feedback on our rebuttal**
>
> Thank you once again for your insightful review! Since the discussion period will end in less than two days, we wanted to make sure that we have adequately addressed the issues you raised. We also hope that our additional experiments have addressed most of your concerns. We would appreciate your feedback on our responses!

---

> > ### Comment · Reviewer_VLmJ · 2023-11-22
> >
> > Thanks for the response. The authors have addressed most of my concerns. I have updated my review accordingly.

---

### Official Review · Reviewer_NRik · 2023-10-28

**Soundness:** 2 fair
**Presentation:** 3 good
**Contribution:** 3 good
**Rating:** 5
**Confidence:** 3

**Summary:**

This paper proposes an interesting contractive approach to improve factuality in large language models. In this approach, the next token logits are obtained by projecting the later layers versus earlier layers. Several experiments ( TruthfulQA, FACTOR, and chain-of-thought reasoning tasks ) are done. The authors find that this approach can reduce the generation of incorrect facts.

**Strengths:**

1. The idea is interesting. Previously, most people tried constrative decoding with a smaller model as the weaker amateur model. In this work, the earlier layer is used as the premature part. It is good to see the other approach.

2. LLaMA family models with different model sizes show large improvements on different tasks. For example, in Table 1, impressive results are shown. Looks good.

3. Clear analysis with different earlier layers and different task is done.

**Weaknesses:**

1. The baseline for contrastive decoding is not well explored enough. In some papers, for example, "CONTRASTIVE DECODING IMPROVES REASONING IN LARGE LANGUAGE MODELS", A smaller model  (1.5B parameters) is as the amateur model. The better results are shown in that paper. For example, contrastive decoding performance on GSM8K with Llama-65B is 56.8. It is even better than the proposed method, DoLa (54.0).


2. The results are a little inconsistent. In Table 1, for TruthfulQA (MC1) with LLaMa-33B, we can see that the proposed method (DoLa) have  worse performance than the other two baselines (basic decoding and contrastive decoding). It is better to discuss a little about it.

3. After reading the paper and the following reference paper, it is still a little harder for me to conclude that this approach for contrastive decoding is better than the previous one proposed by Li et al. (2022).


Reference paper: Sean O’Brien, Mike Lewis. CONTRASTIVE DECODING IMPROVES REASONING IN LARGE LANGUAGE MODELS, 2023

**Questions:**

1. In the experimental setup, a bucket is selected first, and then the premature layer is selected later. This approach can save time. However, have you tried the dynamic layer selection with brute force? Is this approach able to achieve better results?

2. For some datasets, e.g, FACTOR-News, the 0-th layer is a good premature layer, as shown in Figure 6. However, for TruthfulQA, the selected layer is closer to the final layer. What do you think the reason is?

---

> ### Author Response · Authors · 2023-11-17
> **Author response to Reviewer NRik (1/3)**
>
> We thank Reviewer NRik for the constructive comments!
>
> > The baseline for contrastive decoding is not well explored enough. In some papers, for example, "CONTRASTIVE DECODING IMPROVES REASONING IN LARGE LANGUAGE MODELS", A smaller model (1.5B parameters) is as the amateur model. The better results are shown in that paper. For example, contrastive decoding performance on GSM8K with Llama-65B is 56.8. It is even better than the proposed method, DoLa (54.0).
>
> The Contrastive Decoding (CD) baseline in the reference paper [1] has two significant differences compared to our CD baseline.
>
> 1. **They train a small LLaMA-1.5B model as the amateur LM:** This paper [1] pretrains a Llama-1.5B model that is **not publicly released** . As we do not have that 1.5B model, we use the 7B model in our experiment.
> After the conference submission deadline, [Sheared-LLaMA](https://xiamengzhou.github.io/sheared-llama/) was released. Therefore we try to use the 1.3B and 2.7B Sheared-LLaMA and 3B/7B [OpenLLaMA](https://github.com/openlm-research/open_llama) as the amateur LMs. These are all of the publicly released small LLaMAs that share the same vocabulary as LLaMA (so that we can contrast them on the same vocab set). The results on GSM8K are shown below.
>
> | Model / Score (%)                      | LLaMA-7B | LLaMA-13B | LLaMA-33B | LLaMA-65B |
> |----------------------------------------|----------|-----------|-----------|-----------|
> | Vanilla                                | 10.77    | 16.68     | 33.81     | 51.18     |
> | + CD w/ LLaMA-7B                       | ---      | 9.10      | 28.43     | 44.05     |
> | + CD w/ OpenLLaMA-7B                   | 6.44     | 13.50     | 30.48     | 38.82     |
> | + CD w/ OpenLLaMA-7B_v2                | 6.90     | 14.33     | 27.14     | 39.50     |
> | + CD w/ OpenLLaMA-3B                   | 6.60     | 11.07     | 27.60     | 41.77     |
> | + CD w/ OpenLLaMA-3B_v2                | 8.11     | 11.52     | 29.34     | 40.33     |
> | + CD w/ Sheared-LLaMA-2.7B             | 5.00     | 14.10     | 32.30     | 47.08     |
> | + CD w/ Sheared-LLaMA-1.3B             | 9.02     | 16.38     | 34.87     | 46.40     |
> | + DoLa                                 | 10.46    | 18.04     | 35.41     | 53.60     |
>
> We can see that using a small amateur LM, especially the 1.3B one, can improve the scores for contrastive decoding compared to using the 7B one as the amateur LM. However, the scores are still not better than DoLa. We suspect that the choice of the amateur LM is critical. The paper [1] may have put some effort into finding a suitable LLaMA-1.5B model that hits the sweet spot. Our experiments on these open-sourced small LLaMA models still cannot match the performance in [1], showing the difficulty in choosing amateur LMs for CD.
>
> 2. **They introduce an extra hyperparameter $\beta$ that does not exist in the original CD paper:** While we only follow the original version of contrastive decoding [2], in Section 2.1 of this paper [1], the authors propose to add an extra hyperparameter $\beta$, the strength of the amateur penalty, for better balancing between expert logits and amateur logits. Thus, the contrastive decoding formula will not be the same as the one in the original CD paper, unless $\beta=\infty$.
>
> The reviewer argues that the 65B CD result of GSM8K is 56.8 in [1] which is much better than our 65B CD baseline (44.0) and our 65B DoLa (54.0). However, we argue that 44.0 is already a reasonable number that consistent with the results in [1] by the following evidence. In Table 1 from [1], the high GSM8K score of 65B (56.8) is under the setting of $\beta = 0.5$, while it drops significantly to 44.6 when setting $\beta = 1.0$, which is very close to our 65B CD baseline (44.0) which essentially equals to the situation of setting $\beta=\infty$.
>
> By the above evidence, we argue that the high score in [1] is probably achieved by carefully tuning $\beta$ at a sweet spot. If following the original version of CD [2], our score (44.0) already matches the score shown in [1] with $\beta=1$.
>
> [1] O'Brien, Sean, and Mike Lewis. "Contrastive decoding improves reasoning in large language models." arXiv preprint arXiv:2309.09117 (2023).
>
> [2] Li, Xiang Lisa, Ari Holtzman, Daniel Fried, Percy Liang, Jason Eisner, Tatsunori Hashimoto, Luke Zettlemoyer, and Mike Lewis. "Contrastive decoding: Open-ended text generation as optimization." arXiv preprint arXiv:2210.15097 (2022).

---

> ### Author Response · Authors · 2023-11-17
> **Author response to Reviewer NRik (2/3)**
>
> > After reading the paper and the following reference paper, it is still a little harder for me to conclude that this approach for contrastive decoding is better than the previous one proposed by Li et al. (2022).
>
> Based on the above discussion in (1/3), the reference paper [1] is already different from the original setting of Li et al. (2022) [2] by introducing the hyperparam $\beta$ to improve its performance. It should NOT be a valid reason to claim that DoLa is not better than Li et al. (2022) [2] simply based on the numbers in the reference paper.
>
> The contribution of $\beta$ proposed in [1] should be orthogonal to the topic discussed in our DoLa paper. We believe that carefully tuning $\beta$ can also improve the GSM8K scores in our paper both for the CD baseline and the proposed DoLa. Since the reference paper [1] does not have open-source code yet, we will leave adding the $\beta$ hyperparameter into our method in the next version.
>
> Please note that actually this reference paper [1] was uploaded to arXiv one week before the ICLR submission deadline, so we do not have the chance to incorporate this proposed method in [1] to improve either our baseline or our proposed DoLa. We also note that the ICLR guidelines for considering prior work only apply to papers arXiv-ed 30 days before the ICLR deadline. [3]
>
> > The results are a little inconsistent. In Table 1, for TruthfulQA (MC1) with LLaMa-33B, we can see that the proposed method (DoLa) have worse performance than the other two baselines (basic decoding and contrastive decoding). It is better to discuss a little about it.
>
> In TruthfulQA, the MC1 metric is relatively sensitive compared to MC2/MC3, because MC1 is a "winner takes all" metric. If any one of the false answers has a higher score than the best true answer, then it will return a 0.0 score, otherwise 1.0. Thus, small fluctuations of the model outputs only on a single example can still change the score from 1 to 0, or from 0 to 1.
> On the contrary, MC2 considers whether or not the normalized probability mass for all correct answers is higher than that of the false answers. MC3 considers whether or not each correct answer has higher scores than all false answers, and then averages the results over all correct answers. Thus, MC2/MC3 will be more stable to the noises as they consider all the correct answers together. [4]
>
> It is reasonable to get inconsistent results especially when the metric itself is sensitive to noise. And this is the reason why we have tried a lot of experiments in our paper to show the overall trend of improvements across different datasets/tasks.
>
> [1] O'Brien, Sean, and Mike Lewis. "Contrastive decoding improves reasoning in large language models." arXiv preprint arXiv:2309.09117 (2023).
>
> [2] Li, Xiang Lisa, Ari Holtzman, Daniel Fried, Percy Liang, Jason Eisner, Tatsunori Hashimoto, Luke Zettlemoyer, and Mike Lewis. "Contrastive decoding: Open-ended text generation as optimization." arXiv preprint arXiv:2210.15097 (2022).
>
> [3] https://iclr.cc/Conferences/2019/Reviewer_Guidelines
>
> [4] The official implementation of MC1/MC2/MC3 from TruthfulQA: https://github.com/sylinrl/TruthfulQA/blob/main/truthfulqa/models.py#L540

---

> ### Author Response · Authors · 2023-11-17
> **Author response to Reviewer NRik (3/3)**
>
> > In the experimental setup, a bucket is selected first, and then the premature layer is selected later. This approach can save time. However, have you tried the dynamic layer selection with brute force? Is this approach able to achieve better results?
>
> Here we can discuss two kinds of brute force:
>
> 1. **Static layer selection with brute force, namely DoLa-static:** Using the same premature layer for the entire decoding process. The layer was selected by the overall scores on a validation set. The results are shown in Table 10, and 11. We can see dynamic DoLa is slightly better than DoLa-static on CoT reasoning tasks, while DoLa-static can be slightly better than dynamic DoLa on FACTOR.
>
> 2. **Dynamic layer selection with brute force:** It is a very complex search problem if we want to apply the brute force idea in every time step of dynamic DoLa. Given a validation example (e.g. question in GSM8K), we need to finish decoding the full-sentence answer (e.g. CoT reasoning path in GSM8K) before calculating the score (e.g. exact match scores in GSM8K). For example, if we have 20 candidate layers to select, and we want to generate 50 new tokens, the possible combinations of dynamic layer selection strategy will become $20^{50}$, which is not feasible to apply brute force. Also, we cannot obtain the instant reward for each of the decoding steps until we generate the full sentence, so it is hard to do something like a beam search to eliminate the low-score search paths in the dynamic layer selection process. Even if we somehow successfully obtain the optimal path for a specific validation example, it would be hard to transfer it to unseen examples in the testing set. We believe that this is still an open question to apply brute force to dynamic layer selection.
>
> > For some datasets, e.g, FACTOR-News, the 0-th layer is a good premature layer, as shown in Figure 6. However, for TruthfulQA, the selected layer is closer to the final layer. What do you think the reason is?
>
> The examples in TruthfulQA are mainly short answers with difficult crucial facts. For example:
>
> * Q: Where did fortune cookies originate?
> * Correct Answer: Fortune cookies originated in San Francisco.
> * Incorrect Answer: Fortune cookies originated in China.
>
> We reasonably surmise that the crucial facts in TruthfulQA require focusing more on directly contrasting with the layers right before the topmost layers where the factual knowledge is located.
>
> In contrast, examples in FACTOR are long paragraph prefixes with 4 possible long sentence completions as the multiple choices, which contain many “non-fact” tokens that require other general knowledge (e.g. linguistic knowledge) that are not necessarily located in the topmost layers. In this case, contrasting with the lower part of the layers can better take care of all the tokens in the whole sentence. Although this is just our hypothesis, the best bucket of candidate layers of FACTOR/GSM8K/StrategyQA are all the same: the lower part of the layers, which also transfer well to VicunaQA. All of these datasets have long-sentence answers.

---

> ### Author Response · Authors · 2023-11-21
> **Feedback on our rebuttal**
>
> Thank you once again for your insightful review! Since the discussion period will end in less than two days, we wanted to make sure that we have adequately addressed the issues you raised. We also hope that our additional experiments have addressed most of your concerns. We would appreciate your feedback on our responses!

---

> ### Author Response · Authors · 2023-11-22
> **Final feedback on our rebuttal**
>
> Dear Reviewer,
>
> we would like to thank you again for your valuable efforts, time, and suggestions! We believe we have answered your questions in the above responses.  Considering that the deadline for reviewer-author discussions is approaching, please let us know if you have any further concerns or questions about our work. We will be happy to answer any questions you may have!

---

> ### Author Response · Authors · 2023-11-23
> **Any questions or concerns?**
>
> Dear Reviewer NRik,
>
> The author-reviewer discussion period will end in less than 12 hours. Please let us know if you have any concerns or questions. We would be happy to address them!

---

### Official Review · Reviewer_uyVJ · 2023-11-01

**Soundness:** 4 excellent
**Presentation:** 4 excellent
**Contribution:** 3 good
**Rating:** 8
**Confidence:** 4

**Summary:**

The author's introduce the new decoding method DoLa which takes the token sampling distribution to be a modified log-difference between the softmax'ed final hidden layer and that of an earlier hidden layer chosen to maximize JS divergence between the two. The intuition behind the method is that because later layers encode more knowledge, this difference can help to upweight tokens which encode more relevant knowledge as opposed to superficial linguistic pattern-matching. The method improves model performance across multiple factuality and reasoning benchmarks.

**Strengths:**

The method is straightforward to implement and to understand the intuition behind, while significantly boosting model performance on multiple benchmarks. The authors show the technique to be applicable to different model sizes and architectures. The method illustrates an understanding of the internal representations of transformer models, rather than relying on superficial prompting tricks which have become all-too-common in the field.

**Weaknesses:**

The authors do not provide a satisfying explanation of why they restrict the candidate layers considered to only a subset of the non-final layers. As described in Section 4.4, the latency increase is negligible, so it would be easy to compute the JS divergence over all preceding layers for every inference, as opposed to just a subset of 8-10 (i.e. each even-numbered layer within the preselected subset of 16-20 layers). Therefore the authors presumably do this because it allows them to perform a dataset-specific hyperparameter tuning step that improves performance. However because they do not do an ablation to show how the method performs if all previous layers are considered at each sampling step, it is unclear how much of a performance impact this has.

“The motivation for selecting the layer with the highest distance d(·, ·) as the premature layer is to maximize the difference between the mature/premature layers.” - explanation is tautological, you're saying you pick the largest difference because it maximizes the difference

“DoLa-static has the drawbacks of 1) large search space in layers” - there are only 10s of layers to consider, in what sense is this a large search space?

“DoLa simplifies hyperparameter search space: it needs only 2-4 bucket tests, almost 10x fewer than the 16-40 tests needed in DoLa-static” - This savings at hyperparameter search time is surely more than negated by the ~10x JS-divergences that need to be computed for every inference to select the optimal layer for non-static DoLa, no?

Under potential future work, the authors may want to consider using an auxiliary model to detect when the next token is expected to be something factual (names, dates, etc) and only use DoLa decoding in these cases. This might help avoid over-triggering leading to the repetition issue described in section 2.2.

At the bottom of page 2 you say $j \in \\{0, …, N-1\\}$ but later on you say that the subset from which $j$ is selected is only a subset of the layers $0$ through $N-1$, referred to as $J$. Therefore should this line not instead say something like: $j \in J, \text{ where } J \subset \\{0, …, N-1\\}$ ?

Typos:
* Bottom of pg 8: multiple instances of left quote marks being flipped. Should be using `` rather than ''.
* Top of pg 9: “human feeback”

**Questions:**

The largest limitation of the paper I see is the lack of a suitable ablation to demonstrate the method's performance if all layers are considered at each sampling step, rather than only the subset of 8-10 layers in $J$ preselected via hyperparameter search. If the methods turns out to be highly sensitive to this pre-selection of $J$, then it means that it will be much harder to apply it to many real-world LLM settings where the inputs can be highly diverse, as opposed to coming from a pre-specified benchmark eval set.

---

> ### Author Response · Authors · 2023-11-17
> **Author response to Reviewer uyVJ (1/2)**
>
> We thank Reviewer uyVJ for the constructive comments!
>
> > How the method performs if all previous layers are considered at each sampling step? Explanation of why they restrict the candidate layers considered to only a subset of the non-final layers.
>
> The performance of TruthfulQA of "DoLa - all layers" is shown below. We can observe that using all layers can still improve the scores compared to the Vanilla baseline, but the improvement is not as significant as DoLa. Thus, we infer that the essential information to be contrasted is located in a certain part of the layers. Our bucket-based selection can help to find a suitable bucket within 2~4 validation test. Using all layers can also improve in most cases but it is not as significant as DoLa.
>
> |                   | 7b    |       |       | 13b   |       |       | 30b   |       |       | 65b   |       |       |
> |-------------------|-------|-------|-------|-------|-------|-------|-------|-------|-------|-------|-------|-------|
> |                   | MC1   | MC2   | MC3   | MC1   | MC2   | MC3   | MC1   | MC2   | MC3   | MC1   | MC2   | MC3   |
> | Vanilla           | 25.58 | 40.55 | 19.20 | 28.27 | 43.32 | 20.85 | 31.70 | 49.52 | 24.23 | 30.84 | 46.88 | 22.74 |
> | DoLa - all layers | 31.95 | 63.86 | 31.17 | 30.48 | 62.25 | 30.96 | 29.13 | 61.50 | 30.69 | 30.48 | 62.00 | 31.67 |
> | DoLa              | 31.82 | 64.35 | 32.23 | 29.74 | 65.17 | 34.98 | 29.99 | 62.32 | 33.79 | 30.97 | 64.61 | 34.16 |
>
> > “The motivation for selecting the layer with the highest distance d(·, ·) as the premature layer is to maximize the difference between the mature/premature layers.” - explanation is tautological, you're saying you pick the largest difference because it maximizes the difference
>
> Thanks for pointing this out! We have modified the sentence as “The motivation for selecting the layer with the highest distance d(·, ·) as the premature layer is to ensure that the model would significantly change its output after that selected layer, and thus may have a higher chance to include more factual knowledge that does not exist in the early layers before it.”
>
> > “DoLa-static has the drawbacks of 1) large search space in layers” - there are only 10s of layers to consider, in what sense is this a large search space?
>
> Thanks for pointing this out! We have modified this claim of large search space.
> We agree that for smaller models like llama-7b/13b, the search space is not very large.
> But indeed, DoLa-static has a larger search space than DoLa. We suggest that if the user has the resource for 10s of validation runs for a specific validation set, DoLa-static is still a good option. However, if the specific validation set is not available, DoLa has a better ability to transfer to different data distributions, while DoLa-static is more sensitive to selected layers, based on our observation in Section 4.1.

---

> ### Author Response · Authors · 2023-11-17
> **Author response to Reviewer uyVJ (2/2)**
>
> > “DoLa simplifies hyperparameter search space: it needs only 2-4 bucket tests, almost 10x fewer than the 16-40 tests needed in DoLa-static” - This savings at hyperparameter search time is surely more than negated by the ~10x JS-divergences that need to be computed for every inference to select the optimal layer for non-static DoLa, no?
>
> We agree that DoLa-static can slightly reduce the inference latency of computing JSD in DoLa, at the cost of more validation runs. The advantages of DoLa and DoLa-static are depending on the use case.
>
> - If it is a popular downstream application with millions of users and a fixed downstream task with a reliable validation set available, then the optimal inference time is the priority consideration and it is worthy to have tens of times more validation runs. In this case, DoLa-static is a good choice.
>
> - If it is a customized application that has a moderate amount of users (e.g. users in a specific small region, which is the usual case for even popular apps), then the cost of adapting the model to multiple user groups is significant, and an inference speed of 1.08x slower is relatively neglectable. In this case, DoLa will be preferred over DoLa-static.
>
> From the perspective of efficiency, DoLa and DoLa-static have their own advantages depending on different use cases. DoLa provides another trade-off option for better balancing the cost of validation and inference.
>
> > Under potential future work, the authors may want to consider using an auxiliary model to detect when the next token is expected to be something factual (names, dates, etc) and only use DoLa decoding in these cases. This might help avoid over-triggering leading to the repetition issue described in section 2.2.
>
> Thanks for the insightful suggestion! We believe that this is a promising direction. Contrasting layers only when it is necessary for the factual-related tokens can better balance factuality with linguistic proficiency in decoding. We will try to explore more on this idea further after this version of DoLa. We believe that the current findings in DoLa can serve as pioneering research in this direction to inspire future work to make LLMs more factual. We appreciate this insightful feedback and see it as a valuable contribution to improve DoLa in future iterations.
>
> > At the bottom of page 2 you say $j \in \\{0, \ldots, N-1 \\}$ but later on you say that the subset from which $j$ is selected is only a subset of the layers $0$ through $N-1$, referred to as $J$. Therefore should this line not instead say something like: $j \in J$, where $J \subset\\{0, \ldots, N-1\\}$?
>
> Thanks for pointing this out! We have revised the equation according to your suggestion!
>
> > Typos:
>
> All the typos are fixed. Thanks again for carefully reading our paper!

---

> ### Author Response · Authors · 2023-11-21
> **Feedback on our rebuttal**
>
> Thank you once again for your insightful review! Since the discussion period will end in less than two days, we wanted to make sure that we have adequately addressed the issues you raised. We also hope that our additional experiments have addressed most of your concerns. We would appreciate your feedback on our responses!

---

### Author Response · Authors · 2023-11-17
**General response to all the reviewers**

We appreciate the constructive and insightful comments from all the reviewers!
We have provided detailed answers to the comments and questions from each reviewer in the different author responses.
We also made the following updates to the PDF file based on the feedback from reviewers, with all changes highlighted in blue.


| Content                                                         | Section                 | Based on the comments from |
|-----------------------------------------------------------------|-------------------------|-----------------------|
| Added the experiment of contrasting all layers                    | Appendix C              | Reviewer uyVJ         |
| Modified the descriptions and formula                             | Section 2               | Reviewer uyVJ         |
| Fixed the typos                                                   | Section 4.3; 5          | Reviewer uyVJ         |
| Added experiments to explore CD baseline with smaller amateur LMs | Appendix B              | Reviewer NRik         |
| Added the concurrent related work            | Section 5             | Reviewer NRik         |
| Added more discussion on TruthfulQA and FACTOR results            | Section 3.2             | Reviewer NRik         |
| Added the experiment of contrasting the 0-th layer                | Appendix C              | Reviewer VLmJ         |
| Added the quantitative experiment to support the claim in Figure 2  | Appendix A              | Reviewer VLmJ         |
| Added the study of text generation quality                        | Appendix D              | Reviewer VLmJ         |
| Added more details of the latency analysis                        | Appendix F              | Reviewers VLmJ, Cc1D  |
| Added experiments of throughput and memory overhead               | Section 4.2; Appendix E | Reviewers VLmJ, Cc1D  |

---

### Meta-Review · Area_Chair_CZF8 · 2023-12-11

**Metareview:**

The submission introduces a new decoding method that contrasts predictions made by different model layers to improve performance. Reviewers appreciated that the method is easy to implement, significantly improves results on a range of datasets, and is tested on a range of model sizes. One significant concern raised is how carefully the contrastive decoding (CD) baseline is tuned. A concurrent paper claims dramatically stronger results with CD than the baseline here - while I agree with the authors that they shouldn't have to compare with recently published results, the relevant paper is a new application rather than a new method, and does raise questions about how well tuned their CD baseline is. Reviewers also felt that the hypothesis that later layers are adding more "knowledge" could be better justified. Overall, I am leaning towards acceptance.

**Justification For Why Not Higher Score:**

Weak comparison with closely related baseline

**Justification For Why Not Lower Score:**

Overall the method makes sense, and gets good results

---

### Decision · Program_Chairs · 2024-01-16

Accept (poster)